# Behavioral Suite Analysis of Self-Supervised Learning in Atari

**Somjit Nath**[1,2,†], **Rishav Rishav**[1,3,†], **Gopeshh Subbaraj**[1,4,†], **Derek Nowrouzezahrai**[1,2,5], **Samira Ebrahimi Kahou**[1,3,5]

`somjit.nath@mail.mcgill.ca`

[1]**Mila-Quebec AI Institute**
[2]**McGill University**
[3]**University of Calgary**
[4]**Université de Montréal**
[5]**Canada CIFAR AI Chair**

[†] Equal contribution

## Abstract

Self-supervised learning (SSL) methods have shown promise in improving representation learning for reinforcement learning (RL), yet their effectiveness across diverse environments remains unclear. In this paper, we systematically evaluate various SSL methods on a broad set of Atari games, categorized according to the Behavioral Suite (bsuite) framework (Osband et al., 2020). By analyzing performance across these distinct categories, we identify which SSL approaches yield the most significant improvements in different types of Atari environments. Our empirical results, supported by comprehensive performance plots, provide insights into the strengths and limitations of SSL methods in deep RL, guiding future research directions in representation learning for reinforcement learning.

## 1 Introduction

Deep reinforcement learning (RL) has achieved remarkable success in various domains, particularly in environments with well-defined state spaces or pixel-based representations (Mnih et al., 2013; Schulman et al., 2017). However, in more complex scenarios—such as environments with distractors (Stone et al., 2021), partial observability, or real-world settings—learning effective state representations becomes crucial for efficient policy learning. While standard deep RL algorithms implicitly learn representations through their primary RL objectives, these representations may not generalize well to challenging environments, limiting their effectiveness.

Self-supervised learning (SSL) methods have emerged as a promising approach to address this challenge by leveraging auxiliary objectives to learn informative representations without explicit supervision. Several SSL approaches have been proposed, including reconstruction-based methods (Mattner et al., 2012; van Hoof et al., 2016; Watter et al., 2015), forward models predicting future states in latent spaces (Gelada et al., 2019; Schwarzer et al., 2021), intrinsically motivated SSL (Zhao et al., 2022), and contrastive learning techniques (van den Oord et al., 2018; Srinivas et al., 2020). Despite these advances, recent studies suggest that SSL methods may not consistently improve RL performance across all tasks (Li et al., 2022). Thus, a systematic evaluation of SSL methods across diverse environments is necessary to better understand their strengths and limitations.

In this paper, we address this gap by conducting a comprehensive empirical evaluation of various SSL methods across a wide range of Atari games. Inspired by the Behavioral Suite (bsuite) frame-

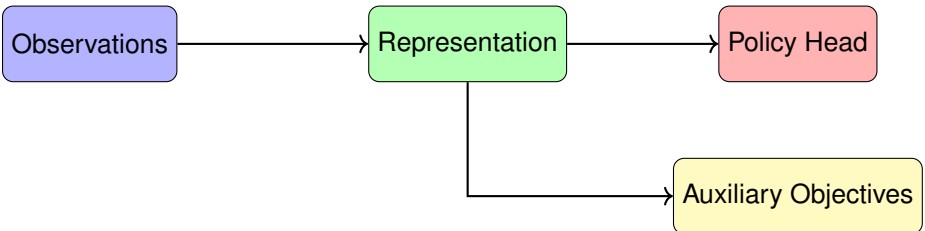

Figure 1: Generic Representation Learning Architecture

work (Osband et al., 2020), we categorize Atari games based on their behavioral characteristics and systematically analyze the performance of different SSL approaches within these categories. Our results, supported by detailed performance plots, provide insights into which SSL methods are most effective in specific types of environments, highlighting their strengths and limitations. This analysis aims to guide future research in representation learning for reinforcement learning, emphasizing the importance of environment-specific SSL method selection.

## 2   Related Work

Deep reinforcement learning (RL) has shown strong performance in environments with well-defined state spaces, such as Atari games, where algorithms like Deep Q-Network (DQN) and Proximal Policy Optimization (PPO) learn policies directly from raw pixel observations (Mnih et al., 2013; Schulman et al., 2017; Hessel et al., 2017b; Haarnoja et al., 2018). These methods implicitly learn representations through end-to-end optimization of the RL objective. However, in environments with distractors (Stone et al., 2021), partial observability, or ambiguous high-dimensional inputs, implicit representations often fail to capture task-relevant information.

To address these limitations, self-supervised learning (SSL) introduces auxiliary objectives (see Figure 1) to learn representations from high-dimensional observations. Broadly, SSL methods in RL can be classified into three categories: predictive models, contrastive learning, and data augmentation techniques. Each category employs a different mechanism to guide representation learning and improve sample efficiency, robustness, and generalization.

Predictive models include reconstruction-based approaches, where autoencoders or variational autoencoders reconstruct the current observation from a latent representation (Mattner et al., 2012; van Hoof et al., 2016; Watter et al., 2015), and forward dynamics models that predict future observations or latent states from the current state and action (Gelada et al., 2019; Schwarzer et al., 2021). These approaches, used in model-based RL methods like World Models, PlaNet, and Dreamer (Ha & Schmidhuber, 2018; Hafner et al., 2018; 2019; 2020), capture temporal dynamics for planning and control. Reward prediction models focus on forecasting immediate rewards from state-action pairs, aligning representations with task-relevant features.

Contrastive learning methods enforce similarity between augmented versions of the same observation and dissimilarity with other observations (van den Oord et al., 2018; Chen et al., 2020; Srinivas et al., 2020). CURL applies this principle in RL by combining contrastive objectives with data augmentation to enhance representation robustness (Srinivas et al., 2020). However, contrastive approaches depend on the availability of diverse negative samples, which can be limited in RL settings, particularly with small datasets (Li et al., 2022).

Data augmentation applies transformations such as cropping, rotation, and jitter to pixel observations, promoting invariance to superficial variations and improving stability (Laskin et al., 2020; Yarats et al., 2021). DrQ integrates augmentations directly into the RL pipeline without requiring auxiliary losses (Laskin et al., 2020), though its effectiveness diminishes in environments with distractors or partial observability (Stone et al., 2021). Despite these approaches, auxiliary SSL objectives do not always improve RL performance, particularly when representation learning and

policy optimization are decoupled (Li et al., 2022). The effectiveness of SSL varies depending on exploration complexity, reward sparsity, and robustness to noise. This paper systematically evaluates SSL methods across Atari environments to clarify their strengths and limitations.

## 3  Behavioral Suite for Atari

### 3.1  Behavioral Suite (bsuite)

The Behavioral Suite (bsuite) (Osband et al., 2020) is a collection of carefully designed reinforcement learning tasks intended to evaluate and analyze the capabilities of RL agents across various dimensions. It categorizes tasks into distinct behavioral challenges, such as exploration, credit assignment, generalization, and robustness, allowing researchers to systematically assess the strengths and weaknesses of RL algorithms.

Inspired by bsuite, we categorize a set of 56 Atari games into distinct behavioral classes to systematically evaluate the effectiveness of self-supervised learning (SSL) methods. This categorization allows us to analyze SSL performance across different types of challenges commonly encountered in RL.

### 3.2  Categorization of Atari Games

We classify each Atari game into one of five categories based on the following logic:

**Basic**: Games where standard Deep Q-Network (DQN) (Mnih et al., 2013) performs better than human-level performance without modifications.
**Credit Assignment**: Games characterized by sparse rewards, requiring effective credit assignment over long time horizons.
**Exploration**: Games with multiple levels or stages that require active exploration to progress.
**Generalization**: Games containing multiple levels with significantly different visual appearances or object configurations, requiring generalization across diverse scenarios.
**Robustness**: All the games are evaluated under varying degrees of reward perturbations, including Gaussian noise (standard deviations: 0.1, 0.3, 1.0, 3.0, 10.0) and reward scaling factors (0.001, 0.03, 1, 30, 1000), to measure robustness to reward signal variations.

Table 1: Categorization of Atari Games based on Behavioral Characteristics

| Category | Atari Games |
|---|---|
| **Basic** | Assault, Asterix, Atlantis, Boxing, Beam Rider, Breakout, Crazy Climber, Defender, Demon Attack, Enduro, Fishing Derby, Freeway, Gopher, Hero, Ice Hockey, Kangaroo, Krull, Kung Fu Master, Ms. Pacman, Name This Game, Phoenix, Pitfall, Pong, Qbert, Road Runner, Robotank, Space Invaders, Star Gunner, Tennis, Time Pilot, Tutankham, Up N Down, Venture, Video Pinball, Yars' Revenge |
| **Credit Assignment** | Gravitar, Hero, Krull, Montezuma's Revenge, Pitfall, Private Eye, Tutankham, Venture |
| **Exploration** | Amidar, Gravitar, Hero, Krull, Montezuma's Revenge, Pitfall, Private Eye, Tutankham, Venture, Zaxxon |
| **Generalization** | Alien, Amidar, Asteroids, Bank Heist, Battle Zone, Centipede, Chopper Command, Double Dunk, Frostbite, Phoenix, Qbert, River Raid, Seaquest, Solaris, Wizard of Wor |

Table 1 summarizes the categorization of the 56 Atari games used in our experiments. This categorization enables us to systematically evaluate and compare SSL methods across distinct behavioral challenges, providing insights into their relative strengths and limitations.

### 3.3 Baseline Methods

We compare the following baselines, each representing a distinct approach to self-supervised learning (SSL) or data augmentation in reinforcement learning:

1. **Rainbow**: Rainbow (Hessel et al., 2017a) serves as our primary baseline and base algorithm upon which improvements are made. Rainbow is a variant of DQN learns value functions directly from raw observations using only the RL loss, without any explicit auxiliary or SSL objectives.

2. **State Reconstruction**: This method augments the Rainbow architecture with an autoencoder. The agent is trained to reconstruct its input observation from a learned latent representation. The reconstruction loss encourages the encoder to capture all information necessary to recreate the input, potentially leading to richer and more informative representations for downstream RL tasks.

3. **Predicting Next State**: Here, the agent is trained with an auxiliary forward dynamics model that predicts the next observation given the current state and action. The prediction loss encourages the learned representation to encode features relevant for modeling environment dynamics, which can be beneficial for planning and generalization.

4. **Predicting Reward**: In this approach, the agent is tasked with predicting the immediate reward from the current state and action as an auxiliary task. This encourages the representation to focus on features that are predictive of reward, potentially improving sample efficiency and credit assignment.

5. **CURL**: Contrastive Unsupervised Representation Learning (CURL) (Srinivas et al., 2020) employs a contrastive loss to learn representations that are invariant to data augmentations. By maximizing agreement between differently augmented views of the same observation and minimizing agreement with other observations, CURL encourages the encoder to capture robust and discriminative features.

6. **SPR**: Self-Predictive Representations (SPR) (Schwarzer et al., 2021) trains the agent to predict its own future latent representations over multiple steps into the future. This temporal consistency objective encourages the representation to capture features that are stable and predictive over time, which can improve both sample efficiency and generalization.

7. **SPR (No Aug)**: This variant of SPR removes data augmentation from the training pipeline, isolating the effect of the self-predictive objective itself. Comparing this to standard SPR highlights the importance of data augmentation in conjunction with temporal consistency objectives.

8. **DrQ**: Data-regularized Q-learning (DrQ) (Kostrikov et al., 2021) improves sample efficiency and robustness by applying strong data augmentations (such as random shifts) to the input images. Unlike other SSL methods, DrQ does not use an explicit auxiliary loss; instead, it relies solely on the RL objective, but with augmented data to encourage invariance and robustness in the learned representations.

## 4 Experiments

To comprehensively evaluate the effectiveness of self-supervised learning (SSL) methods in reinforcement learning, we conduct experiments on a suite of 56 Atari games. Our experimental protocol is designed to assess algorithm performance in both low-data and high-data regimes, as well as to provide robust and statistically meaningful comparisons across diverse game categories.

### 4.1 Experimental Setup

We evaluate each baseline algorithm at two different training durations: **100,000 environment steps (100k)** to represent a low-data regime, and **3.5 million environment steps (3.5M)** to represent a moderate-data regime. This dual setting allows us to analyze how SSL methods impact learning efficiency and performance under varying amounts of available experience.

During training, we periodically evaluate each agent every 10,000 steps. For each evaluation, the agent plays 10 episodes without exploration noise, and we record the average return. This process is repeated throughout the training run, resulting in a sequence of evaluation scores for each algorithm and game.

## 4.2 Score Normalization and Aggregation

To enable fair comparison across games with different reward scales and dynamics, we normalize the performance of each algorithm on each game. Specifically, for each game, we compute the mean evaluation score for each algorithm over the entire training run. We then normalize these scores using min-max normalization:

$$\text{Normalized Score}_{i,g} = \frac{\text{Score}_{i,g} - \min_j \text{Score}_{j,g}}{\max_j \text{Score}_{j,g} - \min_j \text{Score}_{j,g}}$$

where $\text{Score}_{i,g}$ is the mean evaluation score of algorithm $i$ on game $g$, and the minimum and maximum are taken over all algorithms for that game. This normalization ensures that, for each game, the best-performing algorithm receives a score of 1 and the worst receives a score of 0.

## 4.3 Category-wise Analysis

After normalization, we aggregate the scores for each algorithm within each behavioral category as defined in Section 3 (Basic, Credit Assignment, Exploration, Generalization). For each category, we compute the mean and standard deviation of the normalized scores across all games belonging to that category. This allows us to analyze the relative strengths and weaknesses of each SSL method in different types of RL challenges.

For the **Scale** and **Noise** robustness analyses, all 56 games are included, as these criteria are designed to assess the algorithms' robustness to reward perturbations across the entire suite.

## 5 Visualization and Discussion

Figure 2 illustrates the normalized performance of each algorithm across various behavioral categories on 56 Atari games, evaluated at both 100k and 3.5M environment steps. Each experiment was repeated with five independent random seeds to ensure robustness. The per-game rewards, normalized against the Rainbow baseline, are shown in Figure 3. Additionally, aggregate performance metrics based on human-normalized scores, as introduced in Agarwal et al. (2022), are presented in Figure 4.

The results in Figure 2 reveal several important trends regarding the effectiveness of self-supervised learning (SSL) methods:

**Overall Performance**  Most SSL-based methods consistently outperform the standard Rainbow baseline across several behavioral categories at 100k environment steps, highlighting the effectiveness of auxiliary objectives and data augmentation in low-data regimes. However, this advantage becomes much less pronounced at 3.5M steps, where the performance gap between SSL methods and Rainbow narrows considerably.

**Category-wise Trends**

- **Basic:** In the Basic category, SPR (both with and without data augmentation) achieves the highest normalized scores, indicating that temporal consistency and augmentation are especially beneficial in games where Rainbow already performs well. At 3.5M environment steps, however, all algorithms tend to converge in performance, further emphasizing the diminishing advantage of SSL methods as more data becomes available.
- **Credit Assignment:** In the Credit Assignment category, SSL methods that incorporate predictive objectives—such as State Reconstruction, Next State Prediction, and SPR demonstrate clear

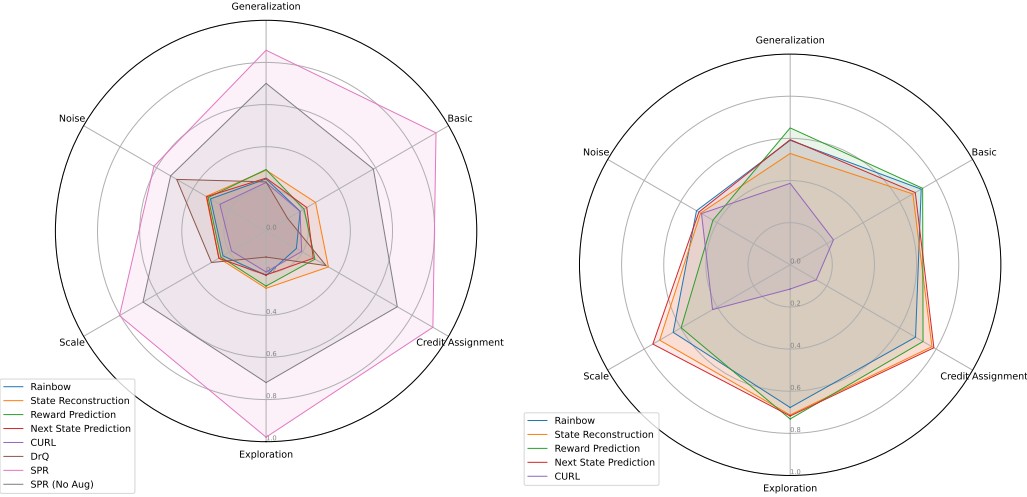

Figure 2: Comparison of algorithm performance across 100k and 3.5M environment steps.

improvements over Rainbow. This suggests that their ability to extract meaningful features from limited experience is particularly beneficial for tasks requiring long-term credit assignment. In contrast, CURL's one-step contrastive loss fails to capture temporal dependencies, limiting its effectiveness in this category. This trend is evident at both 100k and 3.5M environment steps, as shown in Figure 2.

- **Exploration:** In the Exploration category, SPR stands out by leveraging future latent state prediction, enabling the agent to anticipate and seek out novel, informative states. This temporal foresight facilitates more effective exploration compared to methods like CURL, which rely solely on static visual similarity and lack a predictive component. At 3.5M environment steps, we observe that predictive algorithms—similar in nature to SPR—tend to perform particularly well, further underscoring the value of temporal modeling in long-horizon exploration.

- **Generalization:** SPR demonstrates the strongest performance by combining multi-step latent prediction with robust data augmentation, which together enhance the model's ability to generalize across distribution shifts. The variant of SPR without augmentation performs slightly worse, underscoring the critical role of augmentation in achieving robustness. Other predictive methods, such as State Reconstruction and Next State Prediction, also contribute to generalization by explicitly modeling environment dynamics, though they fall short of the consistency achieved by SPR. In contrast, CURL's reliance on one-step contrastive loss limits it to capturing shallow visual invariances without temporal context, resulting in the weakest generalization performance.

**Robustness to Scale and Noise** Across all games, under both Scale and Noise criteria, all algorithms consistently maintain robust performance. However, at higher environment steps, reward prediction methods appear to struggle, potentially due to overfitting to noise or attempting to model abnormally scaled reward signals, which may hinder their generalization and stability.

**Effect of Data Augmentation** Comparing SPR with and without data augmentation demonstrates a consistent advantage for the augmented variant, underscoring the importance of augmentation in low-data settings. Additionally, in low-data regimes, DrQ demonstrates strong robustness in both noise resistance and reward scale sensitivity, highlighting its stability in reinforcement learning environments.

**Poor performance of CURL** Interestingly, CURL consistently underperforms relative to other SSL methods across most categories. Contrastive learning typically requires a large and diverse set of negative samples to learn effective representations—something that is difficult to achieve with

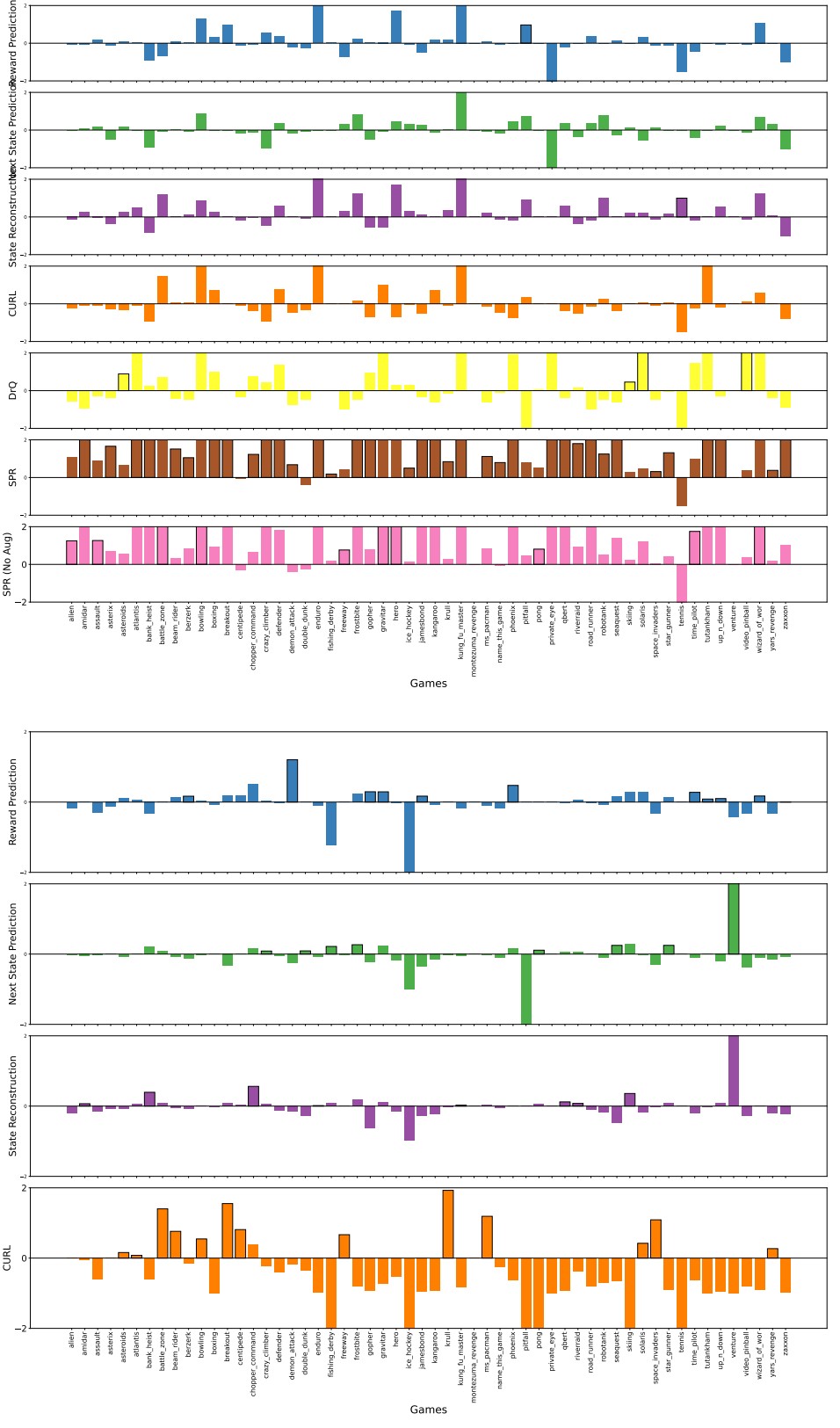

Figure 3: Comparison of algorithm performance per game at 100k and 3.5M steps.

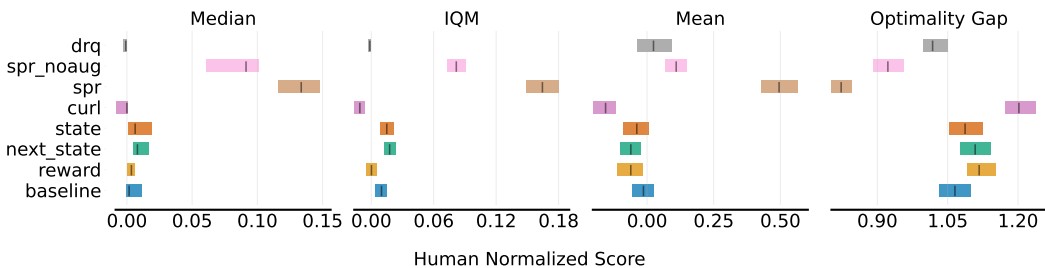

Figure 4: Atari 100k aggregate metrics for all algorithms across 56 Atari games.

limited data. Although, even at higher data scales, CURL continues to lag behind, suggesting that its reliance on shallow visual invariances and lack of temporal modeling may fundamentally limit its effectiveness in more complex reinforcement learning settings.

**Compute Analysis** While SPR delivers strong performance across categories, it is also the most computationally expensive algorithm, requiring significantly more FLOPs per step than others. As shown in Table 2, SPR demands up to 8× more compute than Rainbow. This highlights a key trade-off between performance gains and computational efficiency in SSL-based methods.

| Algorithm | FLOPS per step (GFLOPs) |
|---|---|
| Rainbow | 15 |
| CURL | 25 |
| SPR | 120 |
| SPR No Aug | 90 |
| Reward Prediction | 18 |
| State Reconstruction | 20 |
| Next State Prediction | 21 |

Table 2: FLOPS per step (in GFLOPs) for all algorithms.

In summary, SSL methods, particularly SPR, consistently outperform Rainbow in low-data regimes due to their use of predictive objectives and data augmentation. CURL underperforms across the board, likely due to its lack of temporal modeling and reliance on shallow visual features. Predictive SSL methods are especially effective in settings where temporal structure and long-term dependencies are crucial for learning robust representations.

**Note:** Due to the substantial computational demands, particularly for resource-intensive algorithms like SPR, we have omitted it from our analysis at 3.5M. Additionally, both SPR and DrQ (Kostrikov et al., 2021) were originally designed for the sample efficiency tests on Atari 100k.

## 6    Conclusion

In this work, we conducted a systematic evaluation of self-supervised learning (SSL) methods for representation learning in reinforcement learning, focusing on a diverse set of 56 Atari games categorized according to behavioral criteria inspired by the Behavioral Suite (bsuite). By analyzing performance in both low-data (100k steps) and high-data (3.5M steps) regimes, we provided a comprehensive comparison of several SSL approaches, including predictive, reconstruction, contrastive, and data augmentation-based methods.

Overall, our findings, supported by a detailed bsuite analysis, highlight the importance of selecting appropriate SSL strategies based on the environment and data regime. We hope this study, provides valuable insights for advancing self-supervised representation learning in reinforcement learning research.

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
