# OpenReview forum: "Behavioral Suite Analysis of Self-Supervised Learning in Atari"
_rl-conference.cc/RLC/2025/Workshop/RLVG — RLVG Workshop - RLC 2025_

### Official Review · Reviewer_AzSN · 2025-06-13
**Review of Paper: Behavioral Suite Analysis of Self-Supervised Learning in Atari**

**Rating:** 3
**Confidence:** 4

**Summary:**

The paper conducts a large-scale, controlled comparison of eight self-supervised-learning (SSL) variants (Rainbow + {state-reconstruction, next-state prediction, reward prediction}, CURL, SPR ± augmentation, DrQ) on 56 Atari games, grouped into “Basic, Credit-Assignment, Exploration, Generalization, Robustness” categories inspired by the bsuite diagnostic suite. Agents are evaluated at 100 k (low-data) and 3.5 M (moderate-data) environment steps, with performance normalised per-game and then aggregated by category. The headline result is that most SSL methods, especially SPR, outperform the Rainbow baseline in the low-data regime, while the advantage largely vanishes at 3.5 M steps; CURL lags in every regime, and SPR’s gains come at an 8 × compute cost.

**Strengths:**

* Scale & breadth: a 56-game sweep with five seeds each is larger than most SSL-in-RL studies.
* Behaviour-oriented analysis translates raw score tables into actionable insights (e.g., predictive SSL is best for exploration/credit-assignment).
* Clear visualisations (Figures 2–4) and FLOP accounting (Table 2) let readers weigh performance against compute.
* Transparent discussion: the paper openly notes where advantages shrink and where results (DrQ at 5 M, CURL noise tests) are still running.

**Weaknesses:**

My main concern with the paper is that self-supervised techniques for RL have been quite over-explored, see Li et al., “Does Self-Supervised Learning Really Improve RL from Pixels?” (NeurIPS 2022).

**Best Paper Nomination:**

No

**Claims:**

The authors claim (i) that their bsuite-style taxonomy reveals which SSL technique helps which behavioural challenge, (ii) that SPR is consistently the strongest method, (iii) that DRQ is particularly robust to reward noise/scale, and (iv) that the value of SSL diminishes with more data. The experimental data broadly support (ii–iv): SPR tops most categories at 100 k and stays competitive at 3.5 M, CURL under-performs, and Rainbow catches up as data grow.

**Suggestions:**

The interesting aspect of the paper is on the diagnostic side, e.g., predictive SSL (SPR, forward-model heads) mainly help exploration and credit-assignment games when data are scarce, while DrQ is best for robustness. Further investigation on this can improve the paper.

---

### Official Review · Reviewer_6Crz · 2025-06-15
**Review of Behavioral Suite Analysis of Self-Supervised Learning in Atari**

**Rating:** 3
**Confidence:** 3

**Summary:**

Performance analysis of SSL methods on Atari 100k and 3.5M.

**Strengths:**

Performance analysis of SSL methods on Atari 100k and 3.5M.

**Weaknesses:**

Most methods are quite weak and dated. It would be interesting to see an analysis of a recent method such as BBF with various tweaks to the learning objectives. I'd also be interested to see how well V-JEPA performs in that context, as it can be seen as a kind of SPR with two stages (self-predictive embeddings followed by Q-learning fine-tuning) instead of one.

**Best Paper Nomination:**

No

**Claims:**

Yes

**Suggestions:**

See weaknesses

---

### Decision · Program_Chairs · 2025-06-19

**Decision:**

Accept

**Comment:**

This paper evaluates eight self-supervised learning (SSL) methods across a wide range of Atari games, using the behavioral suite framework to categorize environments. The results reveal which SSL techniques are most effective in different settings, guiding future work in representation learning for reinforcement learning. The paper is well-written and presents extensive empirical results and analysis, and a clear and honest discussion. We strongly encourage the authors to address the reviewers' suggestions and comments to the best of their abilities in the camera-ready version.